# The Nutraceutical N-Palmitoylethanolamide (PEA) Reveals Widespread Molecular Effects Unmasking New Therapeutic Targets in Murine Varicocele

**DOI:** 10.3390/nu13030734

**Published:** 2021-02-25

**Authors:** Pietro Antonuccio, Herbert Ryan Marini, Antonio Micali, Carmelo Romeo, Roberta Granese, Annalisa Retto, Antonia Martino, Salvatore Benvenga, Salvatore Cuzzocrea, Daniela Impellizzeri, Rosanna Di Paola, Roberta Fusco, Raimondo Maximilian Cervellione, Letteria Minutoli

**Affiliations:** 1Department of Human Pathology of Adult and Childhood, University of Messina, 98125 Messina, Italy; pietro.antonuccio@unime.it (P.A.); romeo.carmelo@unime.it (C.R.); roberta.granese@unime.it (R.G.); anna-lisa87@hotmail.it (A.R.); antonellamartino84@gmail.com (A.M.); 2Department of Clinical and Experimental Medicine, University of Messina, 98125 Messina, Italy; hrmarini@unime.it (H.R.M.); salvatore.benvenga@unime.it (S.B.); lminutoli@unime.it (L.M.); 3Department of Biomedical and Dental Sciences and Morphofunctional Imaging, University of Messina, 98125 Messina, Italy; antonio.micali@unime.it; 4Department of Chemical, Biological, Pharmaceutical and Environmental Sciences, University of Messina, 98166 Messina, Italy; salvator@unime.it (S.C.); dimpellizzeri@unime.it (D.I.); rfusco@unime.it (R.F.); 5Department of Paediatric Urology, Royal Manchester Children’s Hospital, Oxford Road, Manchester M13 9WL, UK; raimondo.cervellione@cmft.nhs.uk

**Keywords:** varicocele, testis, PPAR, TLR4, PEA-um, nutraceutical, claudin-11, occludin, TGF-β3, p-ERK 1/2, NLRP3

## Abstract

Varicocele is an age-related disease with no current medical treatments positively impacting infertility. Toll-like receptor 4 (TLR4) expression is present in normal testis with an involvement in the immunological reactions. The role of peroxisome proliferator-activated receptor-α (PPAR-α), a nuclear receptor, in fertility is still unclear. N-Palmitoylethanolamide (PEA), an emerging nutraceutical compound present in plants and animal foods, is an endogenous PPAR-α agonist with well-demonstrated anti-inflammatory and analgesics characteristics. In this model of mice varicocele, PPAR-α and TLR4 receptors’ roles were investigated through the administration of ultra-micronized PEA (PEA-um). Male wild-type (WT), PPAR-α knockout (KO), and TLR4 KO mice were used. A group underwent sham operation and administration of vehicle or PEA-um (10 mg/kg i.p.) for 21 days. Another group (WT, PPAR-α KO, and TLR4 KO) underwent surgical varicocele and was treated with vehicle or PEA-um (10 mg/kg i.p.) for 21 days. At the end of treatments, all animals were euthanized. Both operated and contralateral testes were processed for histological and morphometric assessment, for PPAR-α, TLR4, occludin, and claudin-11 immunohistochemistry and for PPAR-α, TLR4, transforming growth factor-beta3 (TGF-β3), phospho-extracellular signal-Regulated-Kinase (p-ERK) 1/2, and nucleotide-binding oligomerization domain-like receptor (NLR) family pyrin domain-containing 3 (NLRP3) Western blot analysis. Collectively, our data showed that administration of PEA-um revealed a key role of PPAR-α and TLR4 in varicocele pathophysiology, unmasking new nutraceutical therapeutic targets for future varicocele research and supporting surgical management of male infertility.

## 1. Introduction

Varicocele is a dilation of the pampiniform plexus within the spermatic cord, leading to infertility, inflammation, and endocrine disruption [1]. Many factors have been involved in varicocele pathophysiology, including venous hypertension, higher intra-testicular temperature, and increased concentration of reactive oxygen species (ROS) [2]. As a consequence, changes in the seminiferous epithelium, sperm cells apoptosis, and endocrine disruption were demonstrated in experimental varicocele [3,4]. An early diagnosis of varicocele is the gold standard to prevent testicular damage, so that immediate surgical procedures can be performed [5]. However, surgery did not show long-lasting, beneficial effects on varicocele-induced infertility. Therefore, several therapeutic medical strategies, which support surgery, have been proposed to reduce testicular damage and counteract human infertility related to varicocele [6,7].

In this context, there is an intricate relationship between the male genital apparatus and the immune system for reproductive functioning [8]. Immunological reactions induced by the release of cytokines are both risk factors and/or beneficial for male fertility [9]. Toll-like receptors (TLRs) are a group of transmembrane proteins, of which 13 have been identified in mammals [10]. The expression of TLRs has been investigated in epithelial cells of different tissues, such as those of the reproductive tracts. Specifically, TLRs expression was identified in adult human testis [10,11], showing that TLR2, 3 [12], and 4 [13] are highly expressed. Mouse Sertoli cells highly express TLR2, 3, 4, and 5 [14], which can be activated by their agonists and may initiate innate immune responses, inducing increased chemokine monocyte chemotactic protein (MCP)-1 and InterCellular Adhesion Molecule (ICAM)-1 expression [15,16]. Furthermore, a recent study demonstrated that pharmacological inhibition of Toll-like receptor 4 (TLR4) downregulates nucleotide-binding oligomerization domain-like receptor (NLR) family pyrin domain-containing 3 (NLRP3) inflammasome. The inhibition of the NLRP3 pathway was obtained by a combined treatment using the specific TLR4 inhibitor and ligustrazine. This was able to reduce NLRP3 levels [17].

Peroxisome proliferator-activated receptors (PPARs) are another family of transcription factors with the ability to modulate the inflammatory responses. They are divided into three members: (I) α; (II) β/δ; (III) γ [18]. PPARs are adopted nuclear receptors activated by ligands, which control important physiological processes such as energy homeostasis, lipid and glucose metabolism, inflammation, and cell proliferation and differentiation [19]. It was demonstrated that attenuated PPARs expression led to an increased expression of proinflammatory cytokines such as Tumor Necrosing Factor-α (TNF-α), and propagation of an inflammatory response [20]. A key role in fertility control of nuclear receptors was suggested in vivo and in vitro [21]. PPARs are highly expressed in the testis, where beta-oxidation of fatty acids and lipid metabolism are essential for testicular functions [22], such as lipid composition of the sperm and steroids synthesis [23]. PPAR-α activation in Sertoli cells can enhance fatty acid beta-oxidation [24]. Accordingly, synthetic molecules, which bind to PPAR-α, showed positive effects in the treatment of infertility [25].

N-Palmitoylethanolamide (PEA), an endogenous compound of living organisms synthesized from precursor phospholipid, is released following exogenous stimuli in order to prevent the excessive propagation of the inflammatory reaction or to inhibit delayed hypersensitivity reactions [26,27]. PEA is included among the Autacoid Local Inflammation Antagonism amides (ALIAmides), which are implicated in several pathophysiological activities, such as convulsion, pain, inflammation, and neurotoxicity [28,29,30,31,32,33]. Additionally, PEA, acting as ligand and agonist of PPAR-α receptors, shows anti-inflammatory and analgesic activity mediated by activation of PPAR-α [34]. In line with these insights, recent studies in PPAR-α knockout (KO) mice confirmed the role of PPAR-α as a crucial mediator of the anti-inflammatory effects of PEA [35]. Experimental studies have demonstrated that some products of natural origin have antioxidant, anti-inflammatory, and anti-apoptotic effects in varicocele [36,37]. Consequently, the characterization of functional foods of natural origin has suggested the possible use of nutraceuticals in varicocele treatment. In this regard, we recently showed that the bioactive compound lycopene, typically present in the traditional Mediterranean Diet (MD), might be considered a novel strategy in managing varicocele and male infertility [37]. Additionally, a healthy diet aimed at reinforcing our immune system by enhancing the action of endogenous molecules, such as ALIAmides, and, in particular, PEA, or the possibility of using PEA in association with other natural antioxidant molecules, could be helpful in counteracting the pathophysiological imbalance of varicocele.

In light of this background, in the present murine model of varicocele, we investigated the role of the PPAR-α and TLR4 receptors through the administration of ultra-micronized PEA (PEA-um), a ligand agonist of PPAR-α receptors. This was investigated with the aim of unmasking new potentially therapeutic nutraceutical targets in the medical management of male infertility.

## 2. Materials and Methods

### 2.1. Ethics

All experiments were authorized by the University of Messina (Research Mobility Grant, protocol 3518, 18 January 2017) and were in line with the new regulations of USA, Europe, Italy (Welfare Assurance No A5594-01; Directive 2010/63; D.Lgs 2014/26 respectively), and the ARRIVE guidelines.

### 2.2. Animals and Surgical Procedures

A total of 120 male mice (CD-1, PPAR-α KO, and TLR4 KO), weighing 25–30 g (Envigo, Milan, Italy), were used. They received food and water ad libitum. After anesthesia with an intraperitoneal (i.p.) ketamine 80–100 mg/kg and xylazine 10–12.5 mg/kg, varicocele was induced in 60 mice, as previously described in rats [38].

### 2.3. Materials

All compounds used in this study, except where specified, were purchased from Sigma-Aldrich Company Ltd. (Milan, Italy). PEA-um was obtained from Epitech Group SpA (Saccolongo, PD, Italy). Carboxymethylcellulose 1.5% *w*/*v* in saline was employed as vehicle.

### 2.4. Experimental Groups

Twenty-eight days after the surgical procedure, animals were divided in the following groups (*n* = 10 animals for each group) and treated for 21 days as follows:

Sham Vehicle wild-type (WT) group (sham operated + vehicle); Sham Vehicle PPAR-α KO group (sham operated + vehicle); Sham Vehicle TLR4 KO group (sham operated + vehicle); Sham PEA-um WT group (sham operated + PEA-um (10 mg/Kg i.p.) [27]); Sham PEA-um PPAR-α KO group (sham operated + PEA-um (10 mg/Kg i.p.)); Sham PEA-um TLR4 KO group (sham operated + PEA-um (10 mg/Kg i.p.)); Varicocele Vehicle WT group; Varicocele Vehicle PPAR-α KO group; Varicocele Vehicle TLR4 KO group; Varicocele PEA-um WT group; Varicocele PEA-um PPAR-α KO group; Varicocele PEA-umTLR4 KO group. At the end of the treatment, animals were euthanized with an intraperitoneal overdose of ketamine and xylazine, and the left testes were harvested for further analyses.

### 2.5. Histological and Morphometric Evaluation

Testes were harvested, fixed, and stained with hematoxylin and eosin (HE) [39,40]. Images were acquired with a Nikon Ci-L (Nikon Instruments, Tokyo, Japan) microscope.

To evaluate the morphological assessment, mean tubular diameter (MTD) was calculated and seminiferous epithelium was evaluated with the Johnsen’s scoring system [39,40].

### 2.6. Immunohistochemical Analysis

Immunohistochemical analysis was performed as already described [41,42]. Sections were incubated overnight with the following: anti-PPAR-α (1:1000, Santa Cruz Biotechnology, Heidelberg, Germany, sc-398394), anti-TRL4 (1:1000, Santa Cruz Biotechnology, Heidelberg, Germany, sc-293072), anti-claudin-11 (1:100, Thermofisher, Oxford, UK), or anti-occludin (1:100; Thermofisher, Oxford, UK). Images were collected using a Leica DM6 (Milan, Italy) microscope. The percentage area of immunoreactivity (described by the number of positive pixels) was reported as a percentage of the total tissue area [43].

### 2.7. Western Blot Analysis

Western blots were performed as described [41]. Primary antibody: anti-PPAR-α (1:1000, Santa Cruz Biotechnology, Heidelberg, Germany, sc-398394), anti-TRL4 (1:1000, Santa Cruz Biotechnology, Heidelberg, Germany, sc-293072), anti-pERK 1/2 (1:1000, Santa Cruz Biotechnology, Heidelberg, Germany, sc-7383), anti-NLRP3 (1:500, Santa Cruz Biotechnology, Heidelberg, Germany, sc-134306), anti-transforming growth factor-beta3 (TGF-β3), (1:1000, Santa Cruz Biotechnology, Heidelberg, Germany, sc-166833), anti-claudin-11 (1:100, Santa Cruz Bio-technology, Heidelberg, Germany, sc-271232), or anti-occludin (1:100, Santa Cruz Biotechnology, Heidelberg, Germany, sc-133255) [44,45].

### 2.8. Statistical Evaluation

All values in the histograms are expressed as mean ± standard error of the mean of N number of animals [46]. A *p*-Value < 0.05 was considered significant. * *p* < 0.05 vs. sham WT, ^#^
*p* < 0.05 vs. vehicle WT, ** *p* < 0.01 vs. sham WT, ^##^
*p* < 0.01 vs. vehicle WT, *** *p* < 0.001 vs. sham WT, ^###^
*p* < 0.001 vs. vehicle WT; ˆ *p* < 0.05 vs. sham PPAR-α, ^&^
*p* < 0.05 vs. vehicle PPAR-α, ˆˆ *p* < 0.01 vs. sham PPAR-α, ^&&^
*p* < 0.01 vs. vehicle PPAR-α, ˆˆˆ *p* < 0.001 vs. sham PPAR-α, ^&&&^
*p* < 0.001 vs. vehicle PPAR-α; ° *p* < 0.05 vs. sham TLR4, ^§^
*p* < 0.05 vs. vehicle TLR4, °° *p* < 0.01 vs. sham TLR4, ^§§^
*p* < 0.01 vs. vehicle TLR4, °°° *p* < 0.001 vs. sham TLR4, ^§§§^
*p* < 0.001 vs. vehicle TLR4.

## 3. Results

### 3.1. Histopathological Evaluation

In WT, PPAR-α KO, and TLR4 KO sham operated mice treated with vehicle or PEA, a normal structure of both the tubular and the extratubular compartments was demonstrated. Therefore, to ensure the data are clear, a single image and a single value were provided as representative of each group (sham operated + vehicle and sham operated + PEA) (Figure 1A–C). MTD and Johnsen’s score were also normal (Figure 1J,K). On the contrary, in WT, PPAR-α KO, TLR4 KO varicocele mice, and mice treated with vehicle, tubules showed a discontinuous epithelium (Figure 1D–F), a reduced MTD, and a low Johnsen’s score (Figure 1J,K). A marked edema was present in the extratubular compartment. In WT varicocele mice treated with PEA-um, significantly larger tubules with elongated spermatids and a reduced edema in the extratubular compartment were observed (Figure 1G). MTD and Johnsen’s score were close to normal (Figure 1J,K). In PPAR-α KO varicocele mice treated with PEA-um, the germinal epithelium was well preserved, even if an evident edema was present in the extratubular compartment (Figure 1H). MTD and Johnsen’s score were also close to normal (Figure 1J,K). TLR4 KO varicocele mice treated with PEA-um showed a normal germinal epithelium with many spermatids and mature spermatozoa; only a mild edema was present in the extratubular compartment (Figure 1I). In addition, in this group of mice, MTD and Johnsen’s score were close to normal (Figure 1J,K).

### 3.2. Genetic Deficiency of PPAR-α and TLR4 KO

To verify the absence of PPAR-α and TLR4 in mice, we performed Western blot and immunohistochemical analyses. Immunohistochemical analysis of PPAR-α staining showed basal expression in WT (Figure 2A,K) and TLR4 KO animals (Figure 2C,K). PPAR-α expression was reduced in WT (Figure 2D,K) and TLR4 KO varicocele animals (Figure 2F,K) treated with vehicle, while PEA-um administration restored PPAR-α expression to basal levels in both groups (Figure 2G,I,K). PPAR-α KO animals did not display any positive staining in all groups (Figure 2B,E,H,K). Western blot analysis showed the same trend (Figure 2J,L).

Immunohistochemical analysis of TLR4 expression showed no positive staining in WT (Figure 3A,K) and PPAR-α KO animals (Figure 3B,K). TLR4 expression was increased in WT (Figure 3D,K) and PPAR-α varicocele animals (Figure 3E,K) treated with vehicle, while PEA-um treatment reduced TLR4 expression in WT mice (Figure 3G,K). PEA-um administered PPAR-α KO mice did not display any significant reduction in TLR4 expression (Figure 3H,K). TLR4 KO animals did not show any positive staining (Figure 3C,F,I,K). Western blot analysis displayed the same trend (Figure 3J,L).

### 3.3. Effects of the Absence of PPAR-α and TLR4 on Tight Junction Proteins Staining

Basal positive staining of claudin-11 and occludin was detected in sham WT (Figure 4A,J and Figure 5A,J), PPAR-α KO (Figure 4B,J and Figure 5B,J), and TLR4 KO (Figure 4C,J and Figure 5C,J) animals. In vehicle-treated varicocele animals, both proteins were reduced in all groups (Figure 4D–F,J and Figure 5D–F,J). PEA-um administration was able to increase claudin-11 and occludin staining in both WT (Figure 4G,J and Figure 5G,J) and TLR4 KO (Figure 4I,J and Figure 5I,J) groups. PPAR-α KO mice did not show any increase in claudin-11 (Figure 4H,J) or occludin (Figure 5H,J) staining after PEA-um administration. Western blot analysis displayed the same trend (Figure 4K,L and Figure 5K,L).

### 3.4. Effects of the Absence of PPAR-α and TLR4 on TGF-β3, p-ERK 1/2, and NLRP3 Expression

Western blot analysis showed no TGF-β3 expression in all sham operated groups, while vehicle treated varicocele animals displayed increased expression. PEA-um treated varicocele WT and TLR4 KO animals showed a significant reduction of TGF-β3 expression. Conversely, PPAR-α KO varicocele mice administered with PEA-um did not show any significant reduction in TGF-β3 (Figure 6A,B). Phospho-extracellular signal-Regulated-Kinase (p-ERK) 1/2 expression was increased in all varicocele vehicle-treated groups, compared to the sham operated mice. Treatment with PEA-um reduced p-ERK ½ expression in WT and TLR4 KO varicocele administered animals, while in PPAR-α KO varicocele mice treated with PEA-um, the reduction in p-ERK ½ expression was abolished (Figure 6C,D). A significant increase in NLRP3 expression was detected in vehicle WT, PPAR-α KO, and TLR4 KO varicocele mice, compared to the sham operated animals of the same groups. In WT and TLR4 KO varicocele animals administered with PEA-um, a substantial reduction of NLRP3 expression was demonstrated. PEA-um treated PPAR-α KO varicocele animals did not show any downregulation in NLRP3 expression (Figure 6E,F).

## 4. Discussion

Varicocele is one of the main causes of male infertility [47]. It produces ROS accumulation, hypoxia, increase in testis temperature, apoptosis, and the release of proinflammatory cytokines [2,3,4]. Among the receptors highly involved in the male human reproductive function, TLR4 and PPAR-α play a crucial role. PEA, as ligand and agonist of PPAR-α receptors, shows anti-inflammatory and analgesic activity mediated by the activation of PPAR-α [34]. The role of PPAR-α as a mediator of the anti-inflammatory effects of PEA was confirmed in PPAR-α KO mice [35].

Currently, PEA is available worldwide as a nutraceutical in different formulations, with or without excipients [48]. The availability of PEA as an (ultra) micronized nutraceutical formulation (PEA-um) and the lack of side effects [48,49,50], therefore, makes it an attractive candidate in preventive care to counteract inflammatory diseases in humans. Moreover, natural compounds, introduced through food, can play an important role in these pathological conditions. Consequently, the potential therapeutic use of PEA has led many researchers to identify other natural sources rich in this compound [51,52]. In fact, PEA has also been found in the seeds of some legumes, in some vegetables, and in milk [53,54,55,56].

The present study, conducted in a murine model of varicocele using KO mice and PEA-um treatment, aimed to unmask new nutraceutical therapeutic targets for future varicocele research that support surgical management of male infertility.

Moreover, our immunohistochemical and Western blot analyses revealed a significant decrease in PPAR-α expression in the testes from WT and TLR4 KO mice subjected to varicocele, when compared to sham groups. Indeed, PEA-um administration increased PPAR-α expression in varicocele WT and TLR4 KO mice. Conversely, varicocele induction increased TLR4 expression in WT and PPAR-α KO mice, compared to sham. PEA-um treatment downregulated TLR4 expression in varicocele WT mice, while in PPAR-α KO mice TLR4 expression was not modified by PEA-um administration. Collectively, this data further demonstrate the involvement of PPAR-α in the impact of PEA-um.

Indeed, varicocele is a multifactorial and poorly understood pathology, which also impairs spermatogenesis [37]. The blood–testis barrier (BTB) is crucial for spermatogenesis, as it divides in two different sections the seminiferous epithelium: the basal and luminal compartments with junctions between adjacent Sertoli cells [57]. Tight and adherent junctions are made by a transmembrane region, formed by occludin, claudin, and N-cadherin, linked to other proteins, such as ZO-1 and catenin, anchored to the cytoskeleton [57,58]. Animals lacking PPAR-α and TLR4 receptors showed reduced expression of claudin-11 and occludin when subjected to varicocele. Treatment with PEA-um was able to increase both transmembrane proteins expression in WT and TLR4 KO mice, while its positive effect was abolished in the PPAR-α KO group. This apparent contrast between Western blot/immunohistochemical data and histopathological images could be related to the observation that Sertoli cell TJs do not need to be fully functional to allow the early restoration of spermatogenesis, particularly at meiotic and post-meiotic stages [59].

Moreover, it is plausible that, on the basis of our results, the effects of PEA-um on the integrity of BTB in mice could also be related to the direct action of PEA-um on BTB permeability. Accordingly, a recent paper from Couch and coworkers showed that PEA was able to strengthen the integrity of the colonic epithelium through different signaling pathways, among which claudins, aquaporins, and G-coupled protein receptors are included [60].

Normally, BTB function is regulated by many factors. Among them, the cytokine transforming growth factor-beta3 (TGF-β3) modulates the Sertoli cell junction’s dynamic by controlling the opening and closing of the junctions between Sertoli cells and the migration of germ cells [1]. TGF-β3 also mediates its effects on tight junctions via the mitogen-activated protein (MAP) kinases pathway [61]. Many factors might induce BTB disruption, which can lead to a loss of integrity of membrane proteins, can interfere with regular spermatogenesis, and can induce the release of germ cells. These factors include ischemia-reperfusion (I/R) injury, heavy metals, and varicocele [36,37,57,61].

In line with our previous reports [57,61], Western blot analysis showed a substantial increase in TGF-β3 and p-ERK 1/2 in WT and KO mice subjected to varicocele. Indeed, PEA-um administration was able to significantly downregulate TGF-β3 and p-ERK 1/2 expression in WT and TLR4 KO animals. Contrarily, PPAR-α KO mice treated with PEA-um continued to show a marked increase in TGF-β3 and p-ERK 1/2 expression.

One of the main inflammatory pathways induced by varicocele is the NLRP3 inflammasome pathway, as recently demonstrated in animal models and in humans [62,63].

Furthermore, varicocele induction led to an increase in NLRP3 expression in WT and KO animals, with a marked reduction in TLR4 KO mice. However, PEA-um treatment significantly reduced its expression in WT and TLR4 KO mice, while it was ineffective on PPAR-α KO mice.

Overall, our results suggest that, in line with our previous study, PEA-um, via a TLR4-dependent pathway, selectively restores the expression of claudin-11 and occludin and reduces the expression of TGF-β3, p-ERK 1/2 and NLRP3 in varicocele mice. This reveals new potentially therapeutic nutraceutical targets in the medical management of male infertility. Curiously, in this context, the role of PPAR-α appears less important, although PEA-um acts as ligand and agonist of PPAR-α receptors. However, translational medicine frequently revealed that a different impact on humans cannot be ruled out, because multiple factors and interactions, including specific and adaptative immunity, cytokines, oxidative stress, as well as type of pathology and diet style, could dramatically influence the natural history of illness and factors associated with male infertility. Of course, additional experiments are required to better define the role of PEA-um and its nutraceutical bioactivity. Compounds that are present in nature, obtained through diet, can therefore play an important role in maintaining everyone’s well-being [52]. The possibility of enhancing the action of endogenous molecules, such as ALIAmides, and in particular PEA, through a healthy diet, could be a valid alternative to the use of anti-inflammatory drugs, or a possible therapy that works in synergy with the latter. Moreover, the possibility of using PEA in association with other natural antioxidant molecules (e.g., flavonoids, myo-inositol) and/or trace minerals (e.g., selenium) was recently found to increase its therapeutic efficacy [37,52,64].

## 5. Conclusions

In conclusion, our data suggest that the use of PEA might be considered a novel strategy, complementary to surgery, for the treatment of varicocele. It also highlights the importance of the role of PPAR-α and TLR4 in mediating the inflammatory events induced by varicocele.

However, additional and translational studies are required to explore a new possible mechanism of action of this bioactive compound, which is typically present in foods associated with a healthy diet, such as the traditional Mediterranean Diet, in order to elucidate its curative effects in the medical management of varicocele and male infertility.

## Figures and Tables

**Figure 1 nutrients-13-00734-f001:**
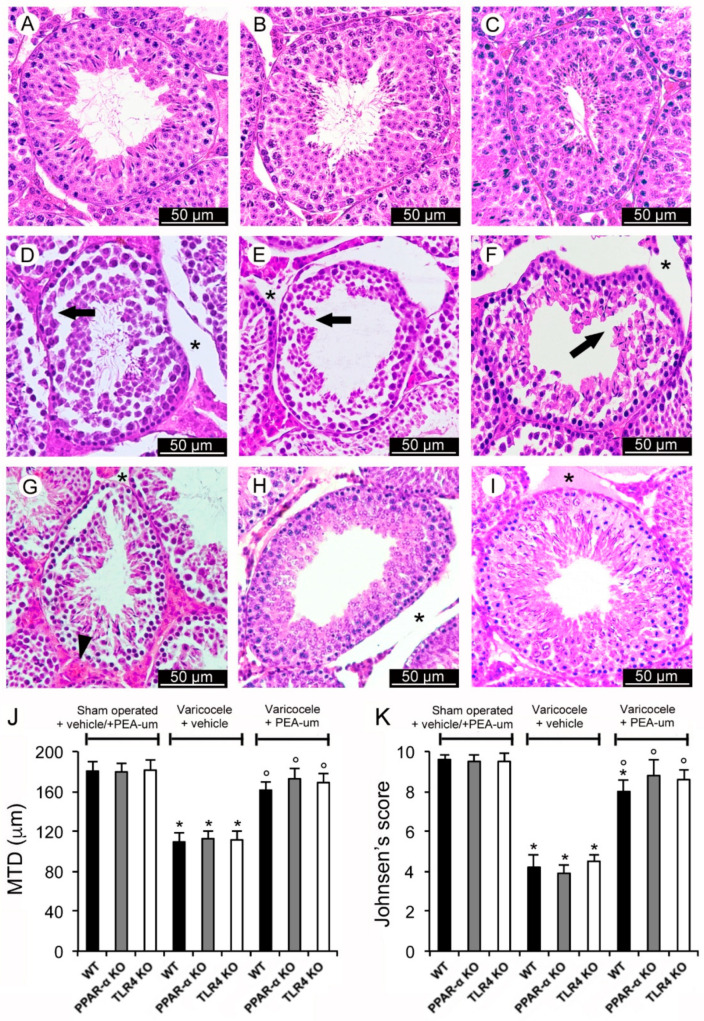
Histological organization of the testes with Hematoxylin-Eosin stain. (**A**–**C**): The images and the means ± SE of only one of the control groups (sham operated + vehicle and sham operated + N-Palmitoylethanolamide (PEA)) are provided, as to not overcomplicate the figure. A normal structure of both the tubular and the extratubular compartments is evident. (**D**–**F**): Wild-type (WT), peroxisome proliferator-activated receptor-α knockout (PPAR-α KO), toll-like receptor 4 knockout (TLR4 KO), varicocele operated mice treated with vehicle, respectively. Tubules with discontinuous epithelium (arrow) and marked extratubular edema (*) are demonstrated. (**G**): WT varicocele operated mice treated with ultra-micronized PEA (PEA-um). Significantly larger tubules with elongated spermatids can be observed. Extratubular edema (*) is reduced and blood vessels (arrowhead) show mild dilation. (**H**): PPAR-α KO varicocele operated mice treated with PEA-um. The germinal epithelium shows a close to normal organization, even if an evident edema (*) is present in the extratubular compartment. (**I**): TLR4 KO varicocele operated mice treated with PEA-um. The germinal epithelium has a normal structure with many spermatids and mature spermatozoa. A mild edema (*) is present in the extratubular compartment. (**J**): Quantitative evaluation of the mean tubular diameter (MTD) in the different groups of mice. (**K**): Johnsen’s score in the different groups of mice. * *p* < 0.05 vs. sham operated groups; ° *p* < 0.05 vs. varicocele plus vehicle groups. (Scale bar: 50 µm).

**Figure 2 nutrients-13-00734-f002:**
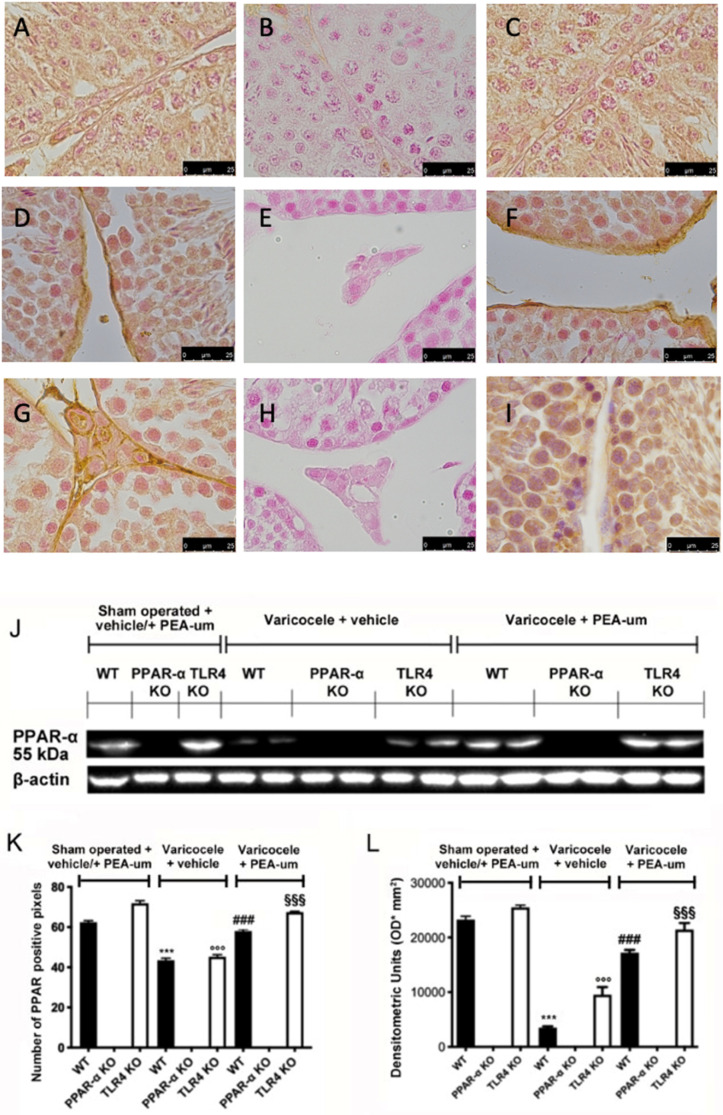
Effects of the absence of PPAR-α and TLR4 on PPAR-α expression. Immunohistochemical evaluation of PPAR-α expression. (**A**): Sham operated WT; (**B**): Sham operated PPAR-α KO; (**C**): Sham operated TLR4 KO; (**D**): Vehicle varicocele WT; (**E**): Vehicle varicocele PPAR-α KO; (**F**): Vehicle varicocele TLR4 KO; (**G**): PEA-um varicocele WT; (**H**): PEA-um varicocele PPAR-α KO; (**I**): PEA-um varicocele TLR4 KO; (**J**): Densitometric analysis; (**K**): Western blot analysis of PPAR-α expression; (**L**): densitometric analysis. Scale bar 100×. *** *p* < 0.001 vs. sham WT, ^###^
*p* < 0.001 vs. vehicle WT; ^°°°^
*p* < 0.001 vs. sham TLR4, ^§§§^
*p* < 0.001 vs. vehicle TLR4.

**Figure 3 nutrients-13-00734-f003:**
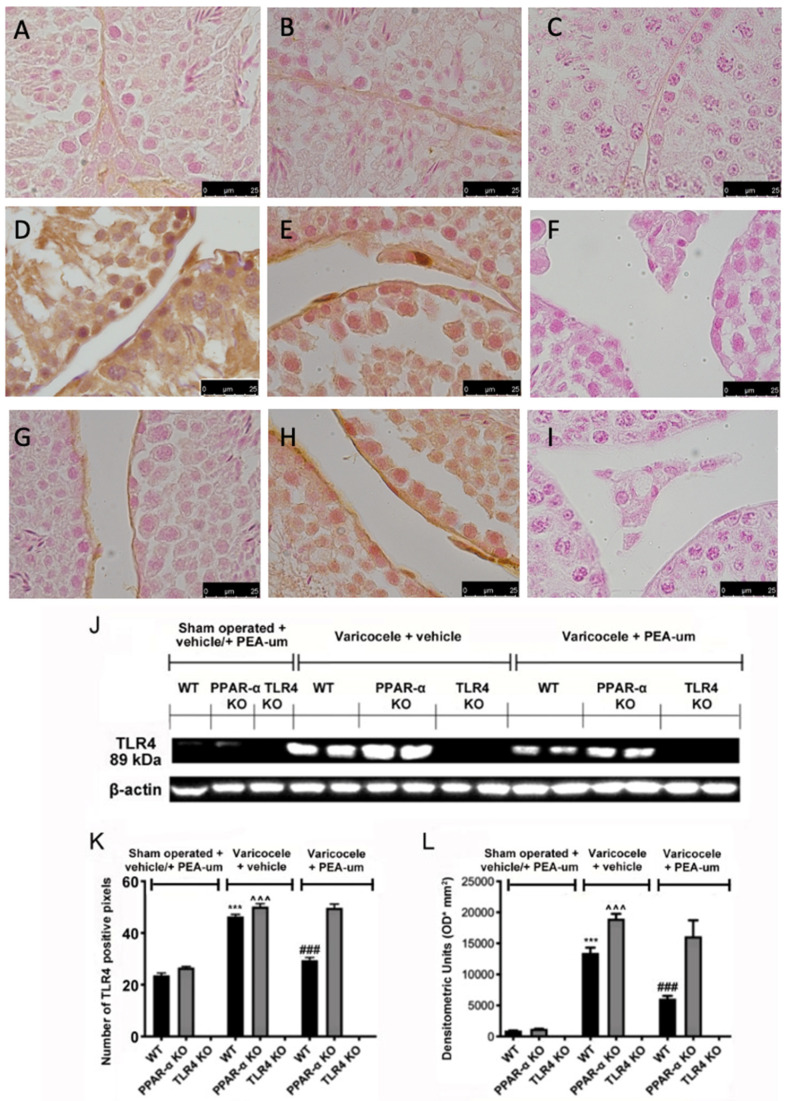
Effects of the absence of PPAR-α and TLR4 on TLR4 expression. Immunohistochemical evaluation of TLR4 expression. (**A**): Sham operated WT; (**B**): Sham operated PPAR-α KO; (**C**): Sham operated TLR4 KO; (**D**): Vehicle varicocele WT; (**E**): Vehicle varicocele PPAR-α KO; (**F**): Vehicle varicocele TLR4 KO; (**G**): PEA-um varicocele WT; (**H**): PEA-um varicocele PPAR-α KO; (**I**): PEA-um varicocele TLR4 KO; (**J**): Densitometric analysis; (**K**): Western blot analysis of TLR4 expression; (**L**): densitometric analysis. Scale bar 100×. *** *p* < 0.001 vs. sham WT, ^###^
*p* < 0.001 vs. vehicle WT ˆˆˆ *p* < 0.001 vs. sham PPAR-α.

**Figure 4 nutrients-13-00734-f004:**
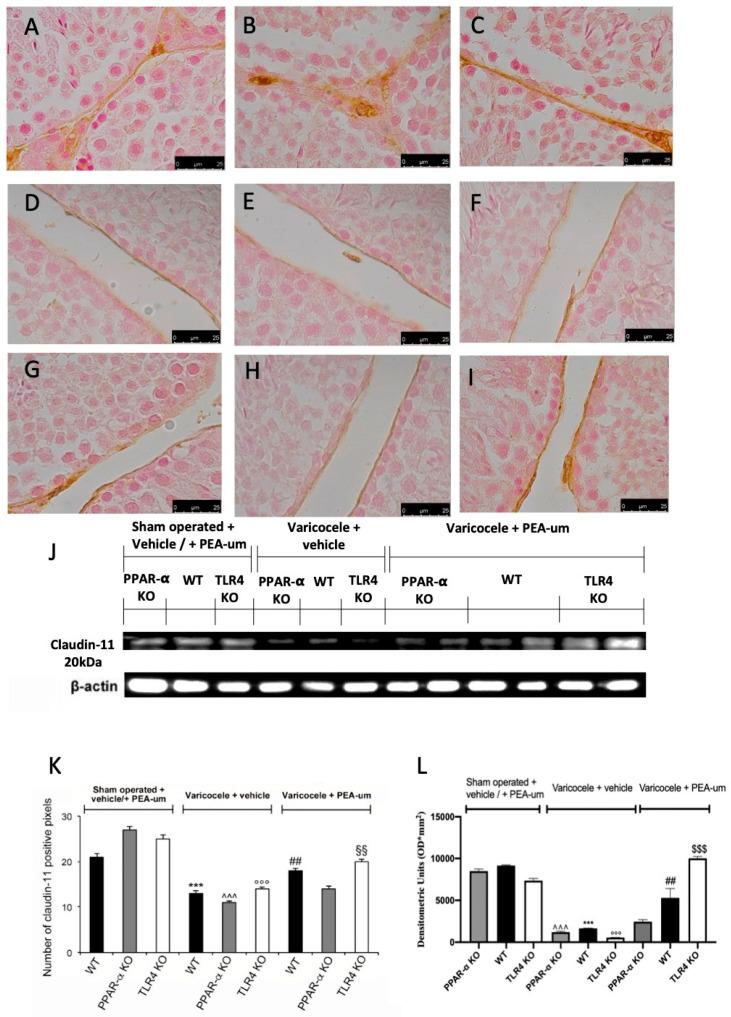
Effects of the absence of PPAR-α and TLR4 on claudin-11 expression. Immunohistochemical evaluation of claudin-11 expression. (**A**): Sham operated WT; (**B**): Sham operated PPAR-α KO; (**C**): Sham operated TLR4 KO; (**D**): Vehicle varicocele WT; (**E**): Vehicle varicocele PPAR-α KO; (**F**): Vehicle varicocele TLR4 KO; (**G**): PEA-um varicocele WT; (**H**): PEA-um varicocele PPAR-α KO; (**I**): PEA-um varicocele TLR4 KO; (**J**): Densitometric analysis. (**K**): Western blot analysis of claudin-11 expression; (**L**): densitometric analysis. Scale bar 100×. ^##^
*p* < 0.01 vs. vehicle WT, *** *p* < 0.001 vs. sham WT, °°° *p* < 0.001 vs. sham TLR4, ˆˆˆ *p* < 0.001 vs. sham PPAR-α, ^§§§^
*p* < 0.001 vs. vehicle TLR4, ^§§^
*p* < 0.01 vs. vehicle TLR4.

**Figure 5 nutrients-13-00734-f005:**
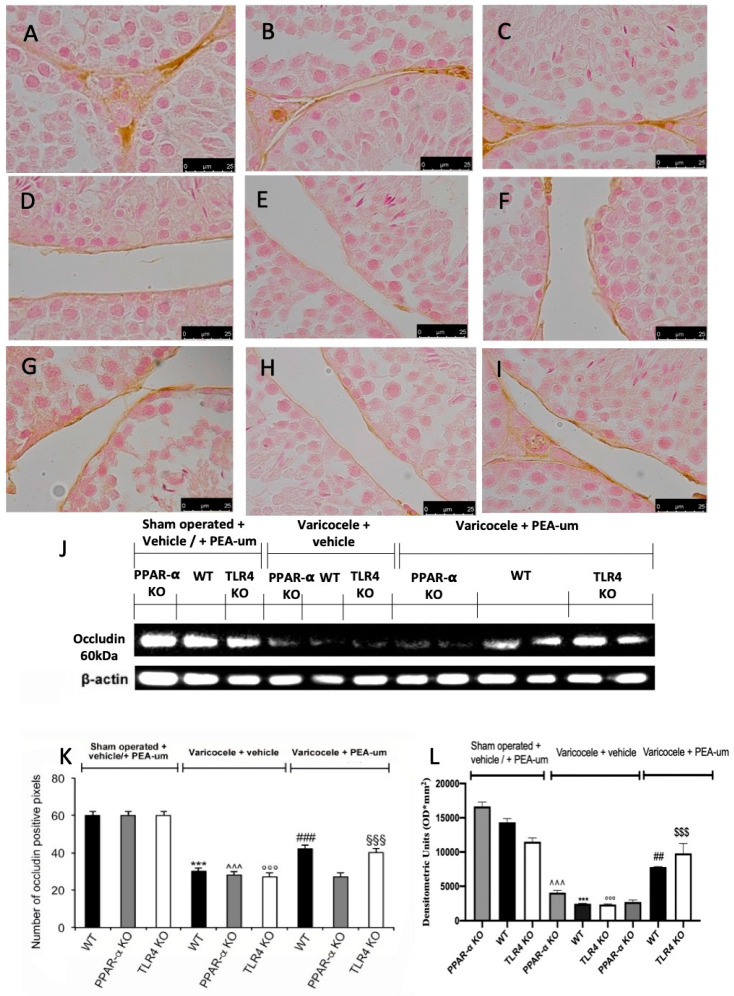
Effects of the absence of PPAR-α and TLR4 on occludin expression. Immunohistochemical evaluation of occludin expression. (**A**): Sham operated WT; (**B**): Sham operated PPAR-α KO; (**C**): Sham operated TLR4 KO; (**D**): Vehicle varicocele WT; (**E**): Vehicle varicocele PPAR-α KO; (**F**): Vehicle varicocele TLR4 KO; (**G**): PEA-um varicocele WT; (**H**): PEA-um varicocele PPAR-α KO; (**I**): PEA-um varicocele TLR4 KO; (**J**): Densitometric analysis. (**K**): Western blot analysis of occludin expression; (**L**): densitometric analysis. Scale bar 100×. *** *p* < 0.001 vs. sham WT, °°° *p* < 0.001 vs. sham TLR4, ˆˆˆ *p* < 0.001 vs. sham PPAR-α, ^§§§^
*p* < 0.001 vs. vehicle TLR4, ^###^
*p* < 0.001 vs. vehicle WT.

**Figure 6 nutrients-13-00734-f006:**
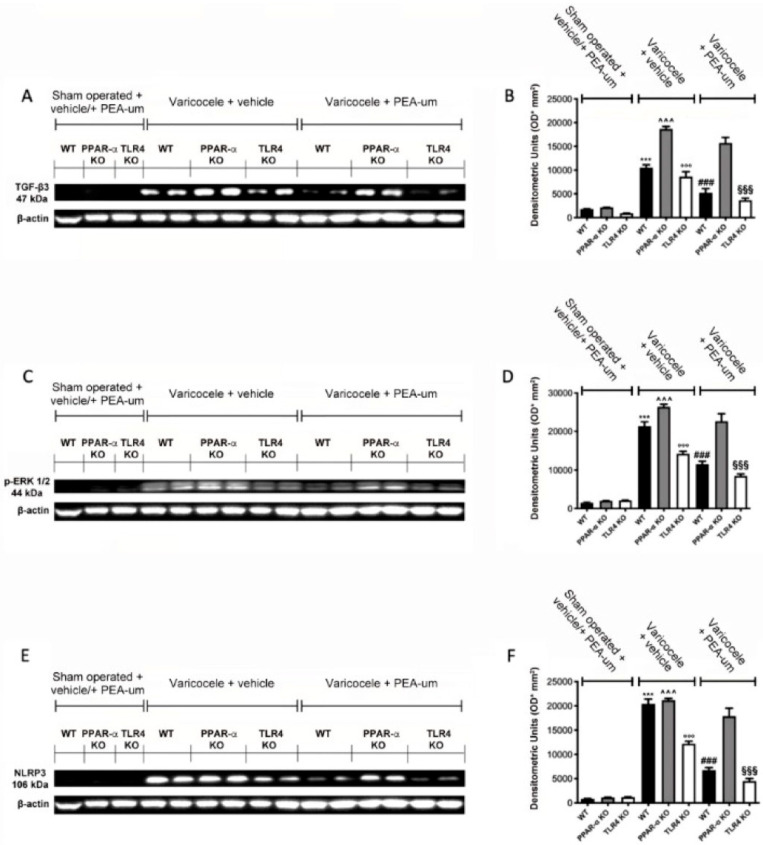
Effects of the absence of PPAR-α and TLR4 on transforming growth factor-beta3 (TGF-β3), p-ERK 1/2, NLRP3 expression. Western blot analysis of TGF-β3 (**A**,**B**), p-ERK 1/2 (**C**,**D**), NLRP3 (**E**,**F**). *** *p* < 0.001 vs. sham WT, °°° *p* < 0.001 vs. sham TLR4, ˆˆˆ *p* < 0.001 vs. sham PPAR-α, §§§ *p* < 0.001 vs. vehicle TLR4, ### *p* < 0.001 vs. vehicle WT. *** *p* < 0.001 vs. sham WT, °°° *p* < 0.001 vs. sham TLR4, ˆˆˆ *p* < 0.001 vs. sham PPAR-α, ^§§§^
*p* < 0.001 vs. vehicle TLR4, ^###^
*p* < 0.001 vs. vehicle WT.

## Data Availability

The data presented in this study are available on request from the corresponding author.

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
