# Peer review of "The Nutraceutical N-Palmitoylethanolamide (PEA) Reveals Widespread Molecular Effects Unmasking New Therapeutic Targets in Murine Varicocele"

_nutrients, 2021, doi:10.3390/nu13030734_

Round 1

Reviewer 1 Report

In this manuscript, the authors investigate the effects of nutraceutical N-Palmitoylethanolamide (PEA) in a mouse model of varicocele. PEA is an endogenous agonist of PPARα receptors, with anti-inflammatory properties. In this study, the role of PPARα and TLR4 receptors was studied through the administration of ultra-micronized PEA (PEA-um). Male wild-type (WT), PPARα KO and TLR4 KO mice were used. A group underwent sham operation and administration of vehicle or PEA-um. Another group (WT; PPARα KO and TLR4 KO) underwent surgical varicocele and was treated with vehicle or PEA-um. The administration of PEA-um revealed a key role of PPARα and TLR4 in varicocele pathophysiology: PEA-um, via a TLR4-dependent pathway, enhances the expression of claudin-11 and occludin and reduces the expression of TGF-β3, p-ERK 1/2 and NLRP3 in varicocele mice. These data suggest that the use of PEA might be considered as a novel strategy, complementary to surgery, for the treatment of varicocele and underline the importance on the role of PPARα and TLR4 in mediating the inflammatory events induced by varicocele.

The study is well designed. The manuscript is clear and well written, and the data presented in this study are novel and interesting. Modifications of the manuscript are required:

  1. Lines 183-185: “To verify that membranes were loaded with equal amounts of protein, they were also incubated with the antibody against laminin (1:1000; Santa Cruz Biotechnology) and GADPH (1:1000; S.C. Biotechnology).”

Please clarify this sentence. The relative protein expression was normalized to β-actin levels. Did the authors also normalize protein expression to laminin and GAPDH levels?

  1. Lines 215-218: “In PPARα KO varicocele mice treated with PEA-um, the germinal epithelium was well preserved, even if an evident edema was present in the extratubular compartment. MTD and Johnsen’s score were also close to normal”.

These results were not discussed. Since PPARα is absent in PPARα KO varicocele mice, how do the authors explain that testicular tissue integrity is better preserved and that MTD and Johnsen’s score are close to normal in PPARα KO varicocele mice treated with PEA-um?

  1. In Figures 4 and 5, the staining and localization of claudin-11 and occludin are hardly visible within the seminiferous tubules at this magnification. Please show images at a higher magnification.
  2. Both Figures 4 and 5 show immunohistochemical analyses of BTB proteins. Please group the results together in a single figure.
  3. Please confirm the effect of PEA-um on the levels of BTB proteins by Western blot analysis.
  4. Lines 321-323: “our results revealed that PEA-um, via a TLR4-dependent pathway, selectively reduces the expression of claudin-11 and occludin, TGF-β3, p-ERK 1/2, NLRP3 in varicocele mice.”

The expression of claudin-11 and occludin is increased when PEA-um is administered. Please correct this sentence.

Author Response

REVIEWER 1

In this manuscript, the authors investigate the effects of nutraceutical N-Palmitoylethanolamide (PEA) in a mouse model of varicocele. PEA is an endogenous agonist of PPARα receptors, with anti-inflammatory properties. In this study, the role of PPARα and TLR4 receptors was studied through the administration of ultra-micronized PEA (PEA-um). Male wild-type (WT), PPARα KO and TLR4 KO mice were used. A group underwent sham operation and administration of vehicle or PEA-um. Another group (WT; PPARα KO and TLR4 KO) underwent surgical varicocele and was treated with vehicle or PEA-um. The administration of PEA-um revealed a key role of PPARα and TLR4 in varicocele pathophysiology: PEA-um, via a TLR4-dependent pathway, enhances the expression of claudin-11 and occludin and reduces the expression of TGF-β3, p-ERK 1/2 and NLRP3 in varicocele mice. These data suggest that the use of PEA might be considered as a novel strategy, complementary to surgery, for the treatment of varicocele and underline the importance on the role of PPARα and TLR4 in mediating the inflammatory events induced by varicocele.

The study is well designed. The manuscript is clear and well written, and the data presented in this study are novel and interesting. Modifications of the manuscript are required:

We thank the Reviewer 1 for her/his suggestions aimed to improve the quality of our paper. First, as requested by the Reviewer 1, the English language and style were revised throughout the manuscript.

  1. Lines 183-185: “To verify that membranes were loaded with equal amounts of protein, they were also incubated with the antibody against laminin (1:1000; Santa Cruz Biotechnology) and GADPH (1:1000; S.C. Biotechnology).” Please clarify this sentence. The relative protein expression was normalized to β-actin levels. Did the authors also normalize protein expression to laminin and GAPDH levels?

- As indicated by the Reviewer, the former Lines 183-185 were changed as follows:

“To verify that membranes were loaded with equal amounts of protein, they were also incubated with the antibody against β-actin (1:1000; Santa Cruz Biotechnology).” We apologize for the mistake.

  1. Lines 215-218: “In PPARα KO varicocele mice treated with PEA-um, the germinal epithelium was well preserved, even if an evident edema was present in the extratubular compartment. MTD and Johnsen’s score were also close to normal”. These results were not discussed. Since PPARα is absent in PPARα KO varicocele mice, how do the authors explain that testicular tissue integrity is better preserved and that MTD and Johnsen’s score are close to normal in PPARα KO varicocele mice treated with PEA-um?

- As indicated by the Reviewer, there is an apparent difference between the morphological data referred to the PPARα KO varicocele mice (well preserved germinal epithelium, MTD and Johnsen’s score close to normal) and the immunohistochemical data on claudin-11 and occluding (no increase in their expression), consequent to the absence of PPARα. However, the analysis of the literature on the recovery of the BTB under experimental conditions demonstrated that Sertoli cell TJs do not need to be fully functional to allow the early restoration of spermatogenesis, particularly at meiotic and post meiotic stages (McCabe et al., 2010), demonstrating the possibility of morphological asynchronous between the close to normal structural organization of the tubules and the immunohistochemical positivity for TJs proteins. In addition, recently it was demonstrated that the expression of occludin could be restored gradually after the administration of bisphenol A (Cao et al., 2020), thus indicating a slow recovery of the BTB, while seminiferous tubules showed a close to normal structure.

To better indicate this apparent difference, the following sentence was added at the former line 300: “This apparent contrast between Western blot/immunohistochemical data and histopathological images could be related to the observation that Sertoli cell TJs do not need to be fully functional to allow the early restoration of spermatogenesis, particularly at meiotic and post meiotic stages (McCabe et al., 2010).”

  1. In Figures 4 and 5, the staining and localization of claudin-11 and occludin are hardly visible within the seminiferous tubules at this magnification. Please show images at a higher magnification.

- As kindly suggested by the Reviewer, we showed Figures 4 and 5 at a higher magnification. Now, the staining and localization of claudin-11 and occludin are clearly visible for readers and researchers.

  1. Both Figures 4 and 5 show immunohistochemical analyses of BTB proteins. Please group the results together in a single figure.

- As kindly suggested by the Reviewer, we tried to add the western blot analysis for each protein expression. However, the images would be too small grouping all results in a single figure. We apologize for the inconvenience.

  1. Please confirm the effect of PEA-um on the levels of BTB proteins by Western blot analysis.

- As kindly suggested by the Reviewer, we performed the Western blot analysis that confirmed data about the effect of PEA-um on the levels of BTB proteins.

  1. Lines 321-323: “our results revealed that PEA-um, via a TLR4-dependent pathway, selectively reduces the expression of claudin-11 and occludin, TGF-β3, p-ERK 1/2, NLRP3 in varicocele mice.” The expression of claudin-11 and occludin is increased when PEA-um is administered. Please correct this sentence.

- As kindly suggested by the Reviewer, the former Lines 321-323 were changed as follows:

“… our results revealed that in varicocele mice PEA-um, via a TLR4-dependent pathway, selectively restores the expression of claudin-11 and occludin and reduces the expression of TGF-β3, p-ERK 1/2 and NLRP3.”

Reviewer 2 Report

In this paper “The Nutraceutical N-Palmitoylethanolamide (PEA) Reveals 2 Widespread Molecular Effects Unmasking New Therapeutic 3 Targets in Murine Varicocele” the author should make a series of necessary corrections before publication. Next I will detail them:

1.-For the measurement of Mean Tubular Diameter (MTD) the author should have made two cross-shaped measurements for each tubular section, to obtain more reliable data

2.-In figure 1G where it says "Extratubular edema is reduced". Refers to interstitial edema and venular blood vessel dilation in the extratubular compartment, point to it and increase the image

3.-Immunohistochemical images (Fig 2, 3, 4 and 5) are not observed well,  there is need to put  images more magnification and better resolution.

4.-Put a magnification bar more appropriate to the image

5.-The author must reveal the role of the interstitial compartment and if he observes changes in it in the expression of PPAR-a, TLR4, claudin-11, occludin that could influence the study.

5.-The author should reveal the interstitial compartment and if he observes changes in the expression of PPAR-a, TLR4, claudin-11, occludin that may influence the study

6.-The discussion should be more elaborate and should show more the results that they have achieved with their work compared to works already done.

Author Response

REVIEWER 2

 We thank the Reviewer 2 for her/his suggestions aimed to improve the quality of our paper. First, as requested by the Reviewer 2, the English language and style were revised throughout the manuscript.

In this paper “The Nutraceutical N-Palmitoylethanolamide (PEA) Reveals 2 Widespread Molecular Effects Unmasking New Therapeutic 3 Targets in Murine Varicocele” the author should make a series of necessary corrections before publication.

Next I will detail them:

1 - For the measurement of Mean Tubular Diameter (MTD) the author should have made two cross-shaped measurements for each tubular section, to obtain more reliable data.

- As requested by the Reviewer, the procedures followed to obtain the Mean Tubular Diameter were better explained by adding in the Materials and Methods section, 2.5 Morphometric evaluation subsection, the following sentence:

“For morphological assessment, the mean tubular diameter (MTD) was calculated by measuring the smallest and largest diameters of 100 separate seminiferous tubules, all showing a circular profile. A Peak Scale Loupe 7x (GWJ Company, Hacienda Heights, USA) micrometer was used as a scale calibration standard to calculate the diameters. The sum of the smallest and largest diameters of each tubule, expressed in μm, was divided by two.”

2 - In figure 1G where it says "Extratubular edema is reduced". Refers to interstitial edema and venular blood vessel dilation in the extratubular compartment, point to it and increase the image.

- As suggested by the Reviewer, Fig. 1G was changed with another micrograph from the same experimental group and both the dilated blood vessels and the reduced extratubular edema were indicated with specific markers. Therefore, the Fig. 1G legend was modified accordingly.

3 - Immunohistochemical images (Fig 2, 3, 4 and 5) are not observed well, there is need to put images more magnification and better resolution.

- We appreciated the reviewer's suggestion and, accordingly, we improved the magnification and put immunohistochemical images with better resolution (Fig 2, 3, 4 and 5).

4 - Put a magnification bar more appropriate to the image.

- We appreciated the reviewer's suggestion and, accordingly, we put a magnification bar more appropriate to the image and added it in the figure legends of the revised manuscript.

5 - The author must reveal the role of the interstitial compartment and if he observes changes in it in the expression of PPAR-a, TLR4, claudin-11, occludin that could influence the study. 6- The discussion should be more elaborate and should show more the results that they have achieved with their work compared to works already done.

- We sincerely appreciated the reviewer's suggestion. As a matter of fact, by improving the magnification of the images to reveal the interstitial compartment of testis, we did not observed any changes in the expression of the proteins that could influence the study. 

Overall, our results suggested, in line with previous study that “in varicocele mice PEA-um, via a TLR4-dependent pathway, selectively restores the expression of claudin-11 and occludin and reduces the expression of TGF-β3, p-ERK 1/2 and NLRP3, then unmasking new potentially therapeutic nutraceutical targets in the medical management of male infertility. Curiously, in this context, the role of PPAR-α appears less important although PEA-um acts as ligand and agonist of PPAR- α receptors. However, translational medicine frequently revealed that it cannot be ruled out a different impact in humans because multiple factors/interactions, including specific and adaptative immunity, cytokines, oxidative stress as well as type of pathology and diet style, could dramatically influence the natural history of illness and male infertility-associated factors. “

We included these comments in the discussion section of revised manuscript (line 321 et seq.).

Round 2

Reviewer 1 Report

The authors addressed most of my recommendations and comments. The manuscript is suitable for publication after revisions:

  1. Lines 215-218: “In PPARα KO varicocele mice treated with PEA-um, the germinal epithelium was well preserved, even if an evident edema was present in the extratubular compartment. MTD and Johnsen’s score were also close to normal”. The authors did not propose any hypothesis to explain why a better testicular tissue integrity, a better MTD and a better Johnsen’s score are obtained in PPARα KO varicocele mice treated with PEA-um. Indeed, since PPARα is absent in these mice, what could be the mechanism of action of PEA-um?
  2. The authors performed Western-blot analyses to confirm the effect of PEA-um on the levels of BTB proteins as requested. However, when looking at the higher magnified images in Figures 4 and 5, the immunolabeling of Claudin-11 and Occludin does not appear convincing: indeed, a uniform (non-specific?) labeling is presented instead of a localization at the basal BTB region within the seminiferous tubules.

Author Response

1. Thanks to the Reviewer 1 for her/his intriguing question. Taking into account that it is quite impossible to evaluate all mechanisms involved in the specific topic, it is plausible that on the basis of our results the effects of PEA-um on the integrity of BTB in mouse could be related to direct action of PEA-um on BTB permeability. Accordingly, a recent paper from Couch and coworkers (2019) showed that PEA was able to strengthen the integrity of the colonic epithelium through different signaling pathways, among which claudins, aquaporins, G-coupled protein receptors are included. Therefore, we hypothesize that, in line with previous papers on testis from our research group, similar mechanisms could be involved in the present experimental model. On the basis of the Reviewer’s suggestion, a specific sentence and a related reference was added in the re-revised manuscript.

2. We thank the Reviewer for her/his acute observation. The submitted images suffered of the poor format quality during the submission, so that some details were lost or were difficult to observe, thus making unconvincing the proposed data. For this reason, we tried to improve the quality of pictures in order to better demonstrate the immunohistochemical behavior of claudin-11 and occludin.

Reviewer 2 Report

All suggested changes have been made. The paper is good for publication but with regard to immunohistochemical images it would be convenient if the author could change the figures: 2D, 2F, 2I, 3D, 3E, 3G, 3H, 4G, 4I, 5I for others of higher quality.

Author Response

We thank the Reviewer for her/his positive evaluation of our paper, whose quality was greatly improved by her/his suggestions. As requested, we tried to improve the figures as indicated, taking into account that submitted images suffered of the poor format quality during the submission, so that some details were lost or were difficult to observe.
